# Recycling and Separation of Homogeneous Catalyst from Aqueous Multicomponent Mixture by Organic Solvent Nanofiltration

**DOI:** 10.3390/membranes11060423

**Published:** 2021-05-31

**Authors:** J.-Kilian Schnoor, Jens Bettmer, Johannes Kamp, Matthias Wessling, Marcel A. Liauw

**Affiliations:** 1Institut für Technische und Makromolekulare Chemie, RWTH Aachen University, Worringerweg 1, 52074 Aachen, Germany; jens.bettmer@rwth-aachen.de; 2Chair of Chemical Process Engineering, Aachener Verfahrenstechnik, RWTH Aachen University, Forckenbeckstrasse 51, 52074 Aachen, Germany; johannes.kamp@avt.rwth-aachen.de (J.K.); matthias.wessling@avt.rwth-aachen.de (M.W.); 3DWI-Leibniz Institute for Interactive Materials, Forckenbeckstr. 50, 52074 Aachen, Germany

**Keywords:** homogeneous catalysis, membranes, nanofiltration, recycling, solvent recovery

## Abstract

Organic solvent nanofiltration (OSN) has evolved to an established recycling method for homogeneous catalysts. However, commercial availability has not circumvented the need for classification and the scoping of possible applications for specific solvent mixtures. Therefore, Evonik’s DuraMem^®^ 300 was assessed for the recycling of magnesium triflate at two transmembrane pressures from a mixture of ethanol, ethyl acetate and water. Catalyst retention up to 98% and permeability of up to 4.44·10^−1^∙L∙bar^−1^∙m^−2^∙h^−1^ were possible when less than 25% ethyl acetate was in the mixture. The retention of some of the components in the ternary mixture was observed while others were enriched, making the membrane also suitable for fractioning thereof.

## 1. Introduction

The massive amounts of flue gases in the steel industry still go unused into waste cleanup. In a recent research consortium (Carbon2Chem, subproject SynAlk), an attempt was made to go from C1 chemistry (CO_x_) to C2 and further to C4 and higher alcohols. While heterogeneous catalysis has been studied for a while, homogeneous catalysis is now also investigated. One of the main challenges is of course the poor separability of the catalyst from the reactants.

In a previous paper, Schnoor et al. have looked at the idea to use organic solvent nanofiltration (OSN) for this separation [1]. During the investigation of the esterification of acetic acid with ethanol with homogeneous catalysts, to produce higher alcohols, different methods were tried for the recycling of the catalysts. Small-scale tests were carried out during which OSN showed good results and was chosen as a feasible option for a larger investigation. In the model system of ethanol, ethyl acetate and cyclohexane, the latter stood for an archetypal organic solvent, and the homogeneous catalyst was separated using OSN. The results were encouraging and were used to adapt the feed composition of the reaction to improve the recyclability of the reaction mixture after the reaction.

In this paper, we have replaced the organic solvent cyclohexane by a much more ecologically friendly solvent, water (H_2_O). Obviously, this is a major change in many respects: dipole moment, hydrogen bond potential, and smaller size to name a few. For the sake of a good comparison, we have repeated many experiments of the previous paper, now with H_2_O instead of cyclohexane as the third component.

The model system is the (continuous) esterification of ethanol and acetic acid for the subsequent hydrogenation to two ethanol molecules [2,3,4]. While in the previous paper the cyclohexane made a second liquid phase, permitting a coupling of reaction and extraction, the H_2_O in this contribution should allow full miscibility with both ethanol and ethyl acetate. Different compositions of the ternary mixture are investigated to identify an optimal process window in which the continuous esterification and a subsequent work-up can be carried out after the reaction. During previous work of Schnoor et al., the results were used to adapt the feed composition of the reaction mixture to meet the optimal parameters needed for the separation of the reaction mixture and the recycling of the catalyst. For the recycling where cyclohexane was replaced by H_2_O, the reaction feed needs further improvement to increase the efficiency of the esterification.

The prediction of the membrane behavior becomes very difficult since it depends on the solvent, the solutes and their affinity to the membrane material. These parameters as well as the size of the molecules, the transmembrane pressure (TMP), the temperature, the feed concentration and the molecules’ charge make the separation with OSN a complex process to predict and to finetune [5,6,7,8,9,10,11,12,13]. Therefore, we had to remeasure the membrane parameters in various ternary solvent mixtures. This way we have determined the catalyst retention as well as the permeation of the membrane under the new conditions.

These results can help other researchers to adopt their processes within the early stages of the process development. In addition, the data provided might help other researchers involved in the integral understanding of the membrane phenomena to further describe these.

As a result, the solvent composition of the reaction mixture shall be adapted to best suit the downstream processing steps for the esterification of acetic acid with short chain alcohols and the hydrogenation thereof. The applicability of OSN is tested, optimal solvent compositions are identified and standard parameters of the membrane are measured. The catalyst retention has to be higher than 90% in a single step to make the implementation of OSN in the continuous process feasible, and has to be able to cope with the reagent stream that is fed in the reactor. OSN allows a wide scalability due to spiral wound membrane modules commercially available in different sizes and enables constant retention and good durability of the membrane.

## 2. Experimental

### 2.1. Materials

The ternary mixture of this work is composed of ethanol, ethyl acetate and H_2_O. An overview of the most relevant properties for the experiments is summarized in Table 1.

During the experiments, the DuraMem^®^ 300 membrane was investigated. The membrane was chosen due to its small molecular weight cut-off (MWCO) but still high transmembrane fluxes, which were determined in different solvent systems during previous work and in literature. The solvent resistant polymeric membrane was created from modified polyimide (PI). The best performance of the membrane was achieved in polar aprotic solvents. The structure of the DuraMem^®^ 300 led to a MWCO of 300 Da. This indicates that 90% of styrene oligomers with a weight of 300 Da as well as other solutes with a molecular weight of 300 g mol^−1^ will be retained by the membrane. The MWCO stated by Evonik Industries was determined through the rejection by the membrane of styrene oligomers dissolved in toluene.

The catalyst retention was determined using the Lewis acid Mg(OTf)_2_ as a model catalyst. The membrane’s MWCO is defined at 300 g mol^−1^ by Evonik Industries; this makes magnesium triflate, with a molecular weight of 322.44 g mol^−1^, one of the less expensive triflate catalysts which can theoretically still be recycled with this membrane.

Figure 1 shows the process scheme of the modified Evonik MetCell^®^ test bench. It consisted of a feed and permeate tank, a high-performance liquid chromatography (HPLC) pump which recycles permeate and pressurizes the whole system, two in-series connected 4″-membrane test cells and a gear pump to circulate the feed.

### 2.2. Experimental Procedure

The cross-flow setup used was a modified Evonik MetCell^®^ test bench. It consisted of two round membranes with an active area of 51 cm^2^ each. In contrast to the stock setup, this one used an HPLC pump to pressurize and circulate the feed into the feed tank. The pressure was set using a freely adjustable back pressure regulator with an overflow into the permeate tank. To avoid concentration polarization, the system was circulated at 60 L h^−1^ using a gear pump. This leads to high flow rates along the membranes and the system as a whole.

Before starting the experiments, the pristine membranes were flushed in an initial cleaning step to wash off production residues. Before the experiment with the catalyst, an additional washing was carried out with the EtOH, H_2_O and EtOAc mixture without the catalyst, to improve the contact between membrane and solvent, remove residual catalyst as well as remaining solvent mixture from the membrane and auxiliary parts of the system. The supporting information provides a more detailed explanation of the necessary steps.

During the first 4 h of the experiment the membranes were flushed in the test cells to reach the steady state, prior to the final pressurization. After pressurization, one permeate sample per test cell and one feed sample were simultaneously drawn every hour for four hours. These four samples were averaged to yield the retention shown in this work. The measurements were carried out in order with increasing pressure, first 30 bar, then 50 bar transmembrane pressure (TMP) to avoid compaction influences of the higher TMP on the experiments at the lower TMP.

### 2.3. Analytics

The retention of the component i was calculated with the concentrations of solutes in the permeate $c_{p,i}$ and the feed $c_{f,i}$; for this they also have to be determined. The retention $R_{i}$ can be calculated with Equation (1).
(1)$$R_{i}=1-\frac{c_{p,i}}{c_{f,i}}$$

According to this formula, a negative retention can be observed with a higher concentration of the solute in the permeate than in the feed. Negative retentions have already been observed and lead to an enrichment of some of the components in the permeate [14,15,16].

^19^F nuclear magnetic resonance (NMR) spectroscopy was used for the quantification of the catalyst. This has been established as a very reliable and robust measurement method during earlier experiments [1].

#### 2.3.1. ^1^H NMR Spectroscopy for Quantitative Analysis of Mixture Composition

The samples needed to be prepared prior to the NMR analysis with the automatization procedure. This was carried out by transferring 400 µL of the sample solution into a NMR tube and adding 100 µL of the lock reference solution. The lock reference was prepared as a 1 mol-% solution of 2,3,4,5,6-pentafluorophenol from abcr in deuterated DMSO (DMSO-d6) from Euriso-top.

The ^1^H-NMR analysis of all experiments was conducted with the Bruker AC 300 from Bruker BioSpin GmbH. The ^1^H-NMR and ^19^F-NMR sequence consisted of the parameters shown in Table 1 in the supporting information. The composition of permeate and feed was determined from ^1^H-NMR spectra of the measured samples. The samples were considered as ternary mixtures due to the very low molar fraction of the catalyst compared to the solvent components (EtOH, H_2_O, EtOAc). This made the analysis of the spectra easier because of the proportionality of number of nuclei contributing to the signal and the integral area of a peak [17].

The molar fraction $x_{i}\;$ of the components can be calculated using Equation (2). $A_{NMR,i}$, the integral area of a certain peak, is divided by $n_{i}$***,*** the number of contributing nuclei of the component ***i***, as well as by the sum of each integral area contributing nuclei ratio.
(2)$$x_{i}=\frac{\frac{A_{NMR,i}}{n_{i}}}{\sum_{i}^{N}\frac{A_{NMR,i}}{n_{i}}}$$

#### 2.3.2. ^19^F NMR Spectroscopy for Quantitative Analysis of Catalyst Concentration

In 2.3.1, the analysis of the samples via ^1^H-NMR measurements using the automatization procedure is described. This required the preparation of the samples as described therein. A Bruker AC 300 from Bruker BioSpin GmbH was used for the ^19^F-NMR analysis of all samples. Table 1 in the supporting information shows the parameters of the ^19^F-NMR sequence. Equation (3) was facilitated to quantify the concentration of the catalyst $c_{\mathrm{cat}}$. For the magnesium triflate, the singlet at −79.11 ppm and for the standard the three signals (q, tq, tt), ranging from −173.99 to −163.34 ppm were used.
(3)$$c_{cat}=\frac{c_{is}\cdot V_{is}\cdot x_{cat}}{V_{s}\cdot x_{is}}\;$$

For the calculation of the catalyst concentration $c_{cat}$ the known values of the concentration of the internal standard $c_{is}$ in mol∙L^−1^, the volume of the internal standard $V_{is}$ in m^3^ and the volume of the sample $V_{s}$ in m^3^ are defined. The change in volume caused by solving pentafluorophenol in DMSO-d6 is negligible. $V_{is}$ can be calculated according to Equation (4).
(4)$$V_{is}=\frac{m_{DMSO-d6}}{\rho_{DMSO-d6}}$$

In Equation (4), the volume of the internal standard $V_{is}$ in m^3^ is calculated with the mass of DMSO-d6 $m_{DMSO-d6}$ in kg and the density of DMSO-d6 $\rho_{DMSO-d6}$ in kg⋅m^−3^. Considering this, Equation (5) is transformed from Equation (3).
(5)$$c_{cat}=\frac{c_{is}\cdot m_{DMSO-d6}\cdot x_{cat}}{V_{s}\cdot x_{is}\cdot \rho_{DMSO-d6}}$$

Using the concentration, the retention of the component ***i***, $R_{i}$ as a percentage, can be calculated with Equation (1).

### 2.4. Determination of the Permeate Flux

With the weight of the samples, the sampling time, and the membrane area, the cross membrane flux J is calculated according to Equation (6). The flux is calculated by considering only the three components in the solvent system.
(6)$$J=\frac{m_{S}}{A\cdot t}\;$$

In this equation, $m_{S}$ is the sample mass in g, total membrane area is A in m^2^ and the time is t in h. The mass fraction $\omega_{i}$ of the component is used to calculate the flux of each solvent component.

## 3. Results and Discussion

In order to evaluate the performance of the DuraMem^®^ 300 for the catalyst recovery in the different solvent mixture compositions, in a first step we discuss the achieved permeate fluxes. Subsequently, we focus on the achieved catalyst retentions. In the last part, we evaluate the retention of each specific solvent in the mixture in dependency on the applied transmembrane pressure.

### 3.1. Permeate Fluxes

The composition of the mixtures significantly influences the permeability during the experiments. The permeability decreases with increasing EtOAc fraction from 4.44 10^−1^⋅L⋅bar^−1^⋅m^−2^⋅h^−1^ at x_EtOAc_ = 0.098 to 1.84·10^−2^ L⋅bar^−1^⋅m^−2^⋅h^−1^ at x_EtOAc_ = 0.741, respectively. The pressure increase from 30 bar to 50 bar TMP increases the total cross-membrane flux. Figure 2 shows the influence of the pressure on the permeability on the right-hand side (yellow triangle).

The best performance of the DuraMem^®^ 300 is reached in polar, more specific polar aprotic solvents such as EtOAc being one. Surprisingly, the performance of the DuraMem^®^ decreases with an increasing fraction of EtOAc, therefore contradicting the product specifications of Evonik. During the preparation of the high-EtOAc fraction solutions, a tendency for the formation of two phases was observed, even though we were still not in the binary region according to literature. We hypothesize that, since EtOH and H_2_O were slightly enriched in the permeate, the EtOAc fraction on the surface of the membrane increased and created a binary mixture by forming small droplets. When these droplets formed on the surface of the membrane, the available area significantly decreased, leading to the decrease of the permeability that we were observing. Unfortunately, it was not possible to prove this, due to the design of the MetCell, but literature reports the blocking of membranes due to binary mixtures [18,19]. Schnoor et al. showed a linear correlation between the EtOAc fraction and the size of the molecules in the mixtures. This size can be described by the hydrodynamic radius [1]. This correlation would explain the reduced permeability of all the components for high molar fractions of EtOAc since large molecules permeate more slowly through the membrane. In alignment with this observation, the permeate fluxes are especially high for the mixtures with high H_2_O ratio.

### 3.2. Catalyst Retention Measurement in Ternary Mixture

For the recycling of magnesium trifluoromethanesulfonate, promising results were demonstrated using the DuraMem^®^ 300 membrane in ternary solvent mixtures [1]. The MWCO of 300 g mol^−1^ of the membrane suggests that the catalyst retention is expected to be higher than 90% since the catalyst has a molecular weight of 322.44 g mol^−1^.

In Figure 2, the retentions are presented for the catalyst at 30 and 50 bar TMP, as well as the retention for EtOAc and the reached permeabilities. In fact, the expected minimal catalyst retention of 90% was not reached for all mixtures, nor for all mixtures of one of the two TMPs, as would be expected from the given MWCO. Observations showed that an increase in TMP resulted in increased retention of some mixtures. The increase in pressure did not appear to significantly increase the retention of the catalyst and the retention of the catalyst was very similar for the different TMPs, with the biggest improvement being 34°%. We assume that this could be caused by the compaction of the membrane by increasing the TMP from 30 bar to 50 bar. As a result, the polymer network was densified, creating smaller effective pores which led to better retention of the catalyst.

Figure 2 shows the reduced retention for the high-EtOAc fraction mixtures. The retention was significantly lower for the mixtures with a high EtOAc fraction. The increased permeation of the catalyst through the membrane for high-EtOAc molar ratios may be caused by the higher affinity of EtOAc to bind to the membrane, according to the solution-diffusion model. This does not necessarily increase the permeation of catalyst through the membrane; we hypothesize that it decreases the overall flux of the different components while the catalyst flux remains constant. This leads to a higher catalyst concentration in the permeate.

### 3.3. Retention Measurement in Ternary Mixture

During the test of OSN as a feasible option for the recovery of the catalyst from the aqueous ternary mixture, the ternary mixture was fractioned in the different components. It was observed that EtOAc was retained, which leads to lower EtOAc concentration, while the concentration of EtOH and especially H_2_O increased in the permeate.

Figure 3 shows the retention of the components in the ternary mixtures at 30 bar and 50°bar TMP. The variation of retention of EtOAc was between −0.47°% and 64.0°%. This resulted in negative retention of EtOH and H_2_O. Considering that EtOAc is the product that should be separated from the other components in this system, OSN is not used any more as a technology to separate the catalyst but for solvent separation. Enriching EtOAc alongside the catalyst creates a mixture that can be used directly for hydrogenation and the Guerbet reaction. This allows a convenient process where EtOH and H_2_O are separated from the mixture at the same time.

Figure 3 illustrates that the composition of the different mixtures has a major influence on the flux of the individual components, leading to large deviations between permeate and retentate composition. On one hand, EtOH is enriched in the permeate up to 25.78% and on the other hand it is retained in the retentate up to 14.20%. This is as well observed for H_2_O which is either enriched in the permeate up to 427.65% or retained in the retentate up to 27.83%. The retention and permeation of EtOAc is also influenced, leading to enrichment in the retentate up to 64.00% or retention in the permeate up to 0.47%. In comparison with Schnoor et al., the separation of the components is smaller and less separation can be achieved in one separation step. The hypothesis is, there is less interaction with the membrane and therefore the differences between the different components govern the separation process [1]. Since the components are similar to each other, less separation can be observed. It can also be observed that the results of the different TMPs are almost identical.

Negative retention is a phenomenon which is still not fully understood and little studied. It has already been observed, e.g., in earlier work of Schnoor et al., Volkov et al., Postel et al. and Marchetti et al. [1,14,15,16]. However, during the experiments it was especially observed for EtOH and H_2_O, which were the smallest and most polar molecules in the solvent mixture. In particular, H_2_O was enriched in the permeate stream because it easily permeates through the membrane. The high H_2_O flux is most likely benefiting from the small molecule size. As a result, solvent mixtures with high H_2_O content show the highest permeability. The high H_2_O content in return can lead to a higher retention of the rather hydrophobic EtOAc due to a high H_2_O content in the membrane, especially on the permeate side. Mixtures with high EtOH content have a mediating effect on the membrane, resulting in little separation of the multicomponent mixture. However, the retention of the catalyst is highest in the mixtures with high EtOH molar ratio.

## 4. Conclusions

The feasibility of using OSN in catalyst recycling from organic solvents was well demonstrated. Magnesium triflate was successfully recycled from an aqueous EtOAc-EtOH-H_2_O ternary mixture by applying OSN to separate the homogeneous catalyst. In addition, the membrane was used to separate the different components. It was able to extract H_2_O from H_2_O-rich mixtures while retaining EtOH and EtOAc in the retentate stream and enriching them. Furthermore, EtOH and H_2_O were extracted from EtOAc-rich mixtures and EtOAc was retained.

From the results, it was possible to identify compositions of the ternary mixture that are well-suited for the recycling process. A permeability of 9.68 10^−2^⋅L⋅bar^−1^⋅m^−2^⋅h^−1^ was reached in a one-step separation with retention of up to 97.47% of the catalyst. A catalyst retention of 90.71% was achieved at the highest permeability of 4.44 10^−1^⋅L⋅bar^−1^⋅m^−2^⋅h^−1^. Those results were derived from the mixtures with the highest EtOH (0.1-0.8-0.1) composition at 30 bar with a feed composition of 13.33 mol-% EtOAc, 69.6 mol-% EtOH and 17.1 mol-% H_2_O, or the highest H_2_O (0.1-0.1-0.8) composition at 30 bar with a composition of the feed of 9.8 mol-% EtOAc, 7.1 mol-% EtOH and 84.5 mol-% H_2_O.

We observed changes in the permeability, the retention of the catalyst and the components in the mixture, which we correlated to the fraction of EtOAc in the ternary mixture. These results provide guidance and allow a precise prediction for future experiments in similar mixtures. However, EtOAc is the desired product for the proposed esterification reaction and even in the aqueous phase a high product concentration is desirable. With the knowledge gained, downstream requirements can be met by adapting the reaction mixtures prior to experiments, when OSN is a feasible option.

## Figures and Tables

**Figure 1 membranes-11-00423-f001:**
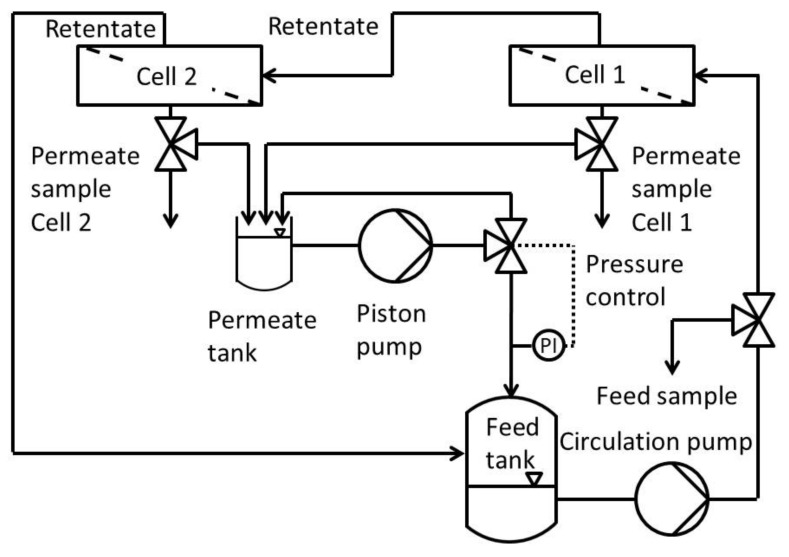
Process scheme of the modified cross flow setup of Evonik MetCell^®^ test bench.

**Figure 2 membranes-11-00423-f002:**
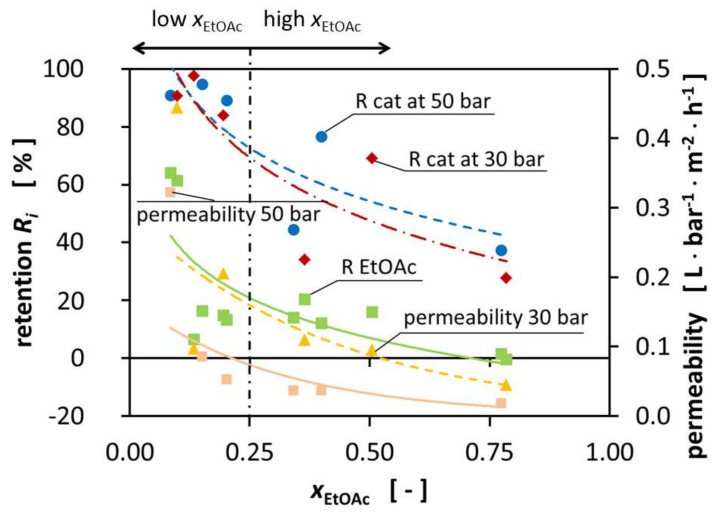
Retention of the catalyst (50 bar TMP blue dot, 30 bar TMP red diamond) and EtOAc (green square) and permeability (30 bar TMP yellow triangle, 50 bar TMP orange square) over the EtOAc molar fraction at room temperature.

**Figure 3 membranes-11-00423-f003:**
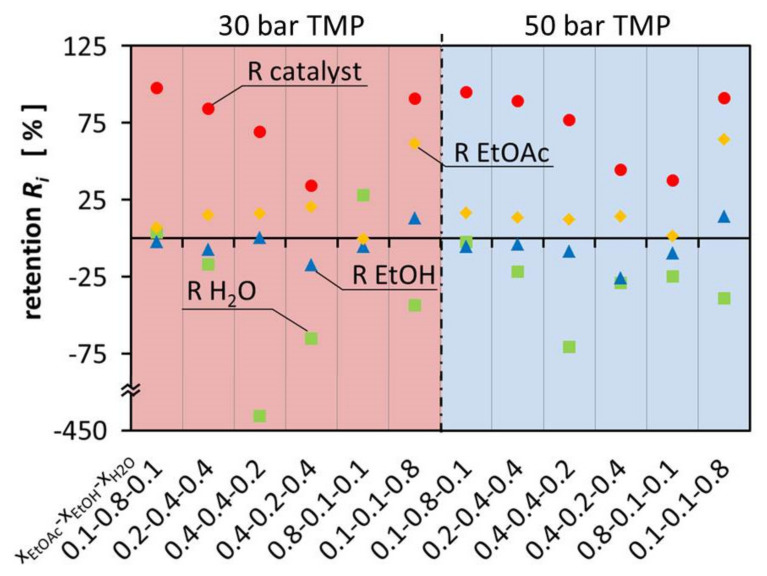
Retention of the components EtOAc, EtOH, H_2_O and the catalyst in aqueous ternary mixture at 30 bar and 50 bar.

**Table 1 membranes-11-00423-t001:** Physicochemical properties of the used solvents.

Solvent	Formula	Mol. Weight [g mol^−1^]	Dyn. Viscosity [mPa s]	Density [kg L^−1^]	Viscosity Blend Number [−]
Ethanol	C_2_H_5_OH	46.07	1.20	0.789	8.468
Ethyl acetate	C_4_H_8_O_2_	88.11	0.44	0.894	−8.810
Water	H_2_O	18.02	1.00	0.99	3.279

## Data Availability

The data presented in this study are available in the supplementary material. Additional data are available on request from the corresponding author(s).

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
