# Peer review of "Recycling and Separation of Homogeneous Catalyst from Aqueous Multicomponent Mixture by Organic Solvent Nanofiltration"

_membranes, 2021, doi:10.3390/membranes11060423_

Round 1

Reviewer 1 Report

This paper studied organic solvent nanofiltration (OSN) which was used to recycle magnesium triflate from an aqueous EtOAc - EtOH- H2O ternary mixture. These results provide guidance and allow a precise prediction for future experiments in similar mixtures. However, there are some problems as following:

1.There are many grammatical mistakes in the paper, like Line 29, 179, 206, and so on.

2.The introduction lacks a comprehensive overview, for example, the description of Schnoor’s work in Line 29.

3.The reference number in the text part should be put before the full stop.

4.There are too many paragraphs in the introduction part, the author should consider combining some paragraphs which have similar meaning.

5.There is a lack of multiplication sign in “4.44 10-1 in Line 173.

6.Figure 2 is suggested to put after the first paragraph of 3.1.

7.In Line 196, why does the catalyst have a higher molecular weight resulting that its retention is expected to be higher than 90%? Please give the explanation in detail.

8.Is there an explanation for the reason that the catalyst retention does not reach 90% for all mixtures in Line 200?

9.There are many hypotheses in the results and discussion part, are there some methods and characterizations to verify the hypotheses?

10.The stability and swelling property of the membranes should be considered.

Reviewer 2 Report

This work by schnoor et al, show recycling and separation of homogenous catalyst by organic solvent Nanofiltration.

The following comments are raised :

  • The paper represents an article; however, it can describe as short communication.
  • Introduction lack the background, problem and novelty.
  • Table .1 is the same in the previous work of the authors
  • Some results in table .2 are adapted from the previous work NMR analysis what’s new?
  • Even figure.3 is similar in design of figure.5 in the previous work which appear in the submitted manuscript in line 247 in the extreme right side. (https://doi.org/10.1002/ceat.201900110)
  • Results and discussion are not described scientific.
  • Most the work is referred to reference 1 which author previous work.

Reviewer 3 Report

The manuscript reports the investigation of the separation efficiency for the catalyst from a mixture  of ethanol, ethyl acetate and water using commercial NF membrane (Duramem(R) 300, which is well written. Authors have been tried to scientifically describe the text and there is no general defects. The results could be helpful to some specific readers studying OSN membranes for recycling valuable products. 

Reviewer 4 Report

See review report attached

Round 2

Reviewer 1 Report

could be accepted in the present form

Author Response

We thank the reviewer for taking the time to read the modified manuscript and giving us the positive feedback. We hope that other researchers can benefit from our findings.

Reviewer 2 Report

Authors respond to all question 

Author Response

(The authors gave the same response as above.)

Reviewer 4 Report

The authors have been addressing most comments and suggestions satisfactorily and improving the manuscript accordingly, making this a better paper. Despite the flaws regarding interpretation of the results, the study is certainly of interest to the OSN and wider membrane community . I have no further comments.

Author Response

(The authors gave the same response as above.)
